# Accurate Global Point Cloud Registration Using GPU-Based Parallel Angular Radon Spectrum

**DOI:** 10.3390/s23208628

**Published:** 2023-10-22

**Authors:** Ernesto Fontana, Dario Lodi Rizzini

**Affiliations:** 1Department of Engineering and Architecture, University of Parma, Parco Area delle Scienze 181/A, 43124 Parma, Italy; dario.lodirizzini@unipr.it; 2Interdepartmental Center for Energy and Environment (CIDEA), University of Parma, Parco Area delle Scienze 95, 43124 Parma, Italy

**Keywords:** registration, mapping, parallel processing, GPU

## Abstract

Accurate robot localization and mapping can be improved through the adoption of globally optimal registration methods, like the Angular Radon Spectrum (ARS). In this paper, we present Cud-ARS, an efficient variant of the ARS algorithm for 2D registration designed for parallel execution of the most computationally expensive steps on Nvidia™ Graphics Processing Units (GPUs). Cud-ARS is able to compute the ARS in parallel blocks, with each associated to a subset of input points. We also propose a global branch-and-bound method for translation estimation. This novel parallel algorithm has been tested on multiple datasets. The proposed method is able to speed up the execution time by two orders of magnitude while obtaining more accurate results in rotation estimation than state-of-the-art correspondence-based algorithms. Our experiments also assess the potential of this novel approach in mapping applications, showing the contribution of GPU programming to efficient solutions of robotic tasks.

## 1. Introduction

In recent years, robot localization and mapping research has been focused on globally optimal registration of point clouds. Registration is the problem of finding the best rigid transformation that links multiple overlapping measurements acquired from different viewpoints. This primitive operation is essential in robot localization, motion tracking systems, and shape reconstruction from partial point clouds. Standard registration methods such as iterative closest point (ICP) [1] are often referred to as local algorithms, as they rely on an accurate initial guess, e.g., provided by robot odometry, to find the transformation that locally minimizes their objective function. Local alignment is usually achieved through associations between corresponding points in the input point clouds. Point matching is feasible when point clouds are close, through a raw assessment of their relative transformation. When a reliable initial estimation is not available, local algorithms may fail to compute consistent and accurate solutions.

Global registration methods [2,3,4,5] compute the aligning pose corresponding to the global minimum of the objective function. These algorithms are often referred to as *certifiable*, as they do not depend (or are less dependent on) an initial guess of the rigid transformation among the input point clouds, and are robust to a large amount of outliers. Global registration usually relies either on robust outlier rejection algorithms for detection of globally consistent correspondences, or on effective global descriptors of point clouds. In order to guarantee global optimality, global registration generally requires computationally intensive operations.

The last decade has been characterized by the rapid and continuous development of graphic cards in terms of performance and application domains. As a matter of fact, the most advanced graphic cards today can be considered additional processing units; they are commonly called Graphic Processing Units (GPUs). As such, global registration methods that are composed of highly parallelizable operations can greatly benefit from GPU processing. In particular, correspondence-less registration methods like Angular Radon Spectrum (ARS) [2,3] separately operate on each point cloud. ARS is a descriptor that captures collinearity among a set of planar points, possibly capturing the simplest and strongest invariant property to rigid motion. The rotation between two point clouds is accurately estimated by finding the maximum correlation between their corresponding spectra. The main limitation of ARS lies in its quadratic complexity, and using GPUs could significantly speed up their computations operating on independent parts. Mathematical frameworks built to deal with similar problems using GPUs have already been proposed [6,7,8], but they tend to lack in guidance for implementation.

In this paper, we propose *Cud-ARS*, a novel parallel algorithm for the registration of planar point clouds using ARS, implemented for execution on Nvidia GPUs through CUDA. ARS descriptors are represented by Fourier coefficients and depended on each point pair. The pairwise assessment of coefficients has been decomposed into independent tasks and assigned to different GPU cores. More specifically, the grid-like structure originates from splitting data into blocks. Computations are then performed among in-block point pairs by parallel threads, then by inter-block comparisons, and finally by a follow-up summation of partial results. In order to limit the computational load on each CUDA thread while also coping with the limited memory capabilities on each said thread, a matrix-like structure, adaptive to the size of the problem to be dealt with, has been implemented and is discussed in this paper. Our self-contained implementation also limits unnecessary dependencies and improves reusability for other projects and frameworks. Cud-ARS has been integrated into a full registration pipeline including translation estimation to perform pairwise scan alignment. Our experiments show a significant reduction in execution time guaranteed by GPU-based assessments of rotation. Moreover, the mapping experiments on standard datasets show that the performance of Cud-ARS pairwise registration is comparable with tools performing state-of-the-art scan-to-map registration. In summary, the contributions of this paper are the following:Introducing the parallel algorithm Cud-ARS for computation of the Fourier coefficients of ARS suitable for parallel execution of GPUs.The implementation of Cud-ARS using the Nvidia CUDA library and conducting experiments comparing performance with state-of-the-art registration methods on benchmarks.A branch-and-bound (B&B)-based translation estimation method improves accuracy over previous versions used in ARS, completing the pose estimation pipeline.

This paper is organized as follows: Section 2 presents the related literature. Section 3 illustrates ARS and its application to point cloud registration. Section 4 presents novel algorithmic contributions, and in Section 5, we discuss our experimental assessment. Finally, Section 6 presents our final remarks.

## 2. Related Work

The scientific literature on registration is extensive, covering several application domains [9] and including different formulations for a variety of problems. Despite it being a problem that has been investigated for over three decades, there is still room for better generalization and overall improvement in areas that are today considered state-of-the-art solutions [10]. A general classification criterion divides registration methods for point clouds into *correspondence-based* and *correspondence-less* methods. Correspondence-based methods rely on the estimation of corresponding points between two point clouds to be aligned. As point association is usually achieved through a rough initial assessment of the transformation between the point clouds, correspondence-based algorithms usually achieve locally optimal registration. Iterative Closest Point (ICP) [1,11] is perhaps the most popular registration algorithm that iteratively refines the transformation by matching each point of the source point set with its closest point in the target set. Notable variants use other cost functions like point-to-plane and plane-to-plane distance [12]. Several approaches have been proposed to address uncertainty in correspondences. The Normal Distributions Transform (NDT) [13,14] performs a soft association based on a probability distribution instead of points. Stein-ICP [15] explicitly evaluates transform uncertainty using Stein variational gradient descent to achieve consistent associations. Recently, Kolpakov and Werman [16] propose an algorithm to assess the initial guess of ICP’s explicit reasoning on point covariances.

Some registration techniques have been proposed to specifically fit widely used sensors like LIDARs. A recent work from Jaimez et al. [17] deals with odometry oriented methods for registration of planar LIDAR data using range flow constraint equations. LIDAR odometry and mapping (LOAM) [18] and its successive variants [19,20] estimate the transformation through detection and association of sparse features like edges and planar patches in point clouds. Customized systems like LIO [21] also integrate other sensory data, such as inertial measurements, in order to improve accuracy.

Correspondence-less and globally optimal methods are less common in the literature. Strong outlier rejection algorithms can make registration less dependent on initial guess or on correspondences. Coherence point drift (CPD) [22] and Vector Field Consensus (VFC) exploit the relative position of points belonging to the same cloud, with the goal of filtering outlier correspondences. TEASER [4] is able to perform registration through a truncated least squares formulation. It also presents sophisticated outlier rejection techniques for dealing with noisy input data. Although they are robust to a high percentage of outliers, these approaches rely on an initial set of correspondences; hence, they cannot be considered *fully* global methods. In contrast, GO-ICP [5] is the clearest example of globally optimal registration using the branch-and-bound method. Although it exploits approximations such as the Euclidean distance transform for closest point computation, this algorithm is computationally intensive and practically unsuitable for online estimation. Another way of addressing correspondence-less registration is by using strong global features in point clouds. The Hough Spectrum (HS) [23,24] and ARS [2,3] are collinearity descriptors for planar point set that can be used in rotation estimation, decoupling it from translation. ARS has the advantages to overcome the discretization issues of HS and to accurately evaluate rotation. Zhang et al. [25] have recently proposed a formulation based on uncertain landmark data. The common element between [25] and ARS stands in basing for the SLAM formulation on working with uncertain data, which should naturally help them in performing better on real sensory data. Other estimation frameworks that are based on the factor graph formulation [26] move in a similar direction. A recent example of this, encompassing an even broader span of applications, is WOLF by Solà et al. [27].

Even if the usage of SLAM methods in real-time applications has been a long-time concern [28,29], the exploitation of GPUs to speed up perception and sensor processing [30] is less frequent in the literature. It is instead more common to see it paired with computer vision primitives or straight-up deep learning methods [31,32,33]. Milioto et al. have proposed Bonnet [34] and RangeNet++ [35] for performing segmentation based on deep learning. Collet et al. [36,37] propose a series of works leading to MOPED, which is a framework for estimating the pose of objects based on recognizing feature keypoints. A typical application for some of these frameworks is robotic manipulation [38]. Furthermore, Titan [39] is a library comprising parallel algorithms to handle geometry in soft-body and multi-robot physics simulations. This approach to Nvidia CUDA parallel processing closely resembles the proposed operational decomposition of ARS. GPU-related literature also includes a class of works that focuses on high-level formal analysis of computational optimization and parallelization. Ha et al. [40] present an optimal parallel scan method, showing experiments on throughput and MIPS on a data-intensive simulation of a prefix sum problem. The goal of this paper is to help fill the gap between parallelization analysis of benchmark problems and deep learning-related applications, specifically in the field of robotic registration and mapping.

## 3. Angular Radon Spectrum for Registration

### 3.1. Angular Radon Spectrum of a Gaussian Mixture Model

ARS is a function suitable for the estimation of rotation between two point clouds. It has been introduced in [2,3]. Input point clouds must be represented as a Gaussian Mixture Model (GMM). Although other solutions are possible, a straightforward choice to convert a point cloud into a GMM is to associate each point to a Gaussian kernel centered on the point, with a covariance matrix representing the uncertainty about the point position. We let $P={\left\{\mu_{i}\right\}}_{i=1,\ldots ,n_{p}}$ with $\mu_{i}\in R^{2}$ be the estimated position vectors of the points. It is convenient to define the *density function* $f:R^{2}\to R_{⩾0}$ that represents the point density in the plane and is proportional to the probability density function (PDF) of finding a point. Then, the GMM density function is defined as
(1)$$\begin{array}{c} \begin{array}{cc} & f\left(r\right)=\sum_{i=1}^{n_{p}}w_{i}\;f_{i}\left(r\right)=\sum_{i=1}^{n_{p}}w_{i}\;n\left(r-\mu_{i},\Sigma_{i}\right), \end{array} \end{array}$$
where the sum of positive weights $w_{i}$ is equal to one, and symbol $n\left(\mu_{i},\Sigma_{i}\right)$ is used for a Gaussian kernel of mean value $\mu_{i}$ and covariance matrix $\Sigma_{i}$. The *Radon Transform* (RT) [41] of f(r) enables measuring the alignment of the point set with a given line $F_{q}$ represented by parameters $q={\left[\theta ,\rho \right]}^{⊤}$. The RT is defined as
(2)$$\begin{array}{cc} R_{}\left[f\right]\left(q\right)\; & =\int_{F_{q}}f\left(r\right)\;dr=\sum_{i=1}^{n_{p}}w_{i}\;\int_{F_{q}}n\left(r-\mu_{i},\Sigma_{i}\right)\;dr\;. \end{array}$$

In our case, the integral of line $F_{q}$ can be solved through the parametric equation of points r(t) lying on line
(3)$$\begin{array}{cc} r(t) & =t_{1}\;u_{1}+t_{2}\;u_{2}=Ut, \end{array}$$
where $t_{1}=\rho$ is a fixed constant, $t_{2}$ is the varying parameter associated to the points on the line, $u_{1}=\hat{u}\left(\theta \right)={\left[\cos\theta ,\sin\theta \right]}^{⊤}$ is the unitary vector orthogonal to the line, and $u_{2}=\hat{u}\left(\theta +\pi /2\right)={\left[-\sin\theta ,\cos\theta \right]}^{⊤}$ corresponds to the line direction. Since r is a linear transformation, the integral of each Gaussian kernel has an elegant closed-form expression
(4)$$R_{}\left[f_{i}\right]\left(\theta ,\rho \right)=n\left(\rho -{\tilde{\mu }}_{i,1},{\tilde{\sigma }}_{i,1}^{2}\right),$$
where ${\tilde{\mu }}_{i,1}=u_{1}^{⊤}\mu_{i}$ and ${\tilde{\sigma }}_{i,1}^{2}=u_{1}^{⊤}\Sigma_{i}u_{1}$.

The ARS is a function applied to RT to detect patterns of points collinear to a given direction measured by θ. Given a superadditive concentration function κ(·), the ARS is defined as
(5)$$S\left[f\right]\left(\theta \right)=\int_{-\infty }^{+\infty }\kappa \left(R_{}\left[f\right]\left(\theta ,\rho \right)\right)\;d\rho .$$

A standard concentration function is $\kappa \left(x\right)=x^{2}$, which is implicitly used in the remaining. Thus, the square of a sum of Gaussian kernels $\kappa (R_{}\left[f\right])$ in Equation (Equation 1) consists of double products of Gaussians that can be integrated. The equation of the ARS of a GMM has the form
(6)$$S\left[f\right]\left(\theta \right)=\sum_{i=1}^{n_{p}}\sum_{j=1}^{n_{p}}w_{i}\;w_{j}\;\psi_{ij}\left(\theta \right)\;.$$

The ARS kernel functions $\psi_{ij}\left(\theta \right)$ are equal to the Gaussian-like function
(7)$$\psi_{ij}\left(\theta \right)=n\left(u_{1}^{⊤}\left(\theta \right)\left(\mu_{i}-\mu_{j}\right),u_{1}^{⊤}\left(\theta \right)\left(\Sigma_{i}+\Sigma_{j}\right)u_{1}\left(\theta \right)\right)\;.$$

The ARS $S\left[f\right]\left(\theta \right)$ is π-periodic and can be expanded into Fourier series. The Fourier series of ARS kernel $\psi_{ij}$ is
(8)$$\psi_{ij}\left(\theta \right)=a_{0}^{\left(ij\right)}+\sum_{k=1}^{+\infty }\left(a_{k}^{\left(ij\right)}\cos\left(2k\theta \right)+b_{k}^{\left(ij\right)}\sin\left(2k\theta \right)\right)\;.$$

In the *isotropic* case, the Fourier coefficients $a_{k}^{\left(ij\right)}$ and $b_{k}^{\left(ij\right)}$ have closed-form equations [2]
(9)$$a_{k}^{\left(ij\right)}=2\;I_{k}\left(\lambda_{ij}\right)\;e^{-\lambda_{ij}}\;{\left(-1\right)}^{k}\;\cos\left(2k\theta_{ij}\right),$$
(10)$$b_{k}^{\left(ij\right)}=2\;I_{k}\left(\lambda_{ij}\right)\;e^{-\lambda_{ij}}\;{\left(-1\right)}^{k}\;\sin\left(2k\theta_{ij}\right),\;$$
where $I_{k}\left(\right)$ is the modified Bessel function of the first kind, $\lambda_{ij}={∥\mu_{i}-\mu_{j}∥}^{2}/\left(8\sigma^{2}\right)$ and $\theta_{ij}=∠\left(\mu_{j}-\mu_{i}\right)$. In the anisotropic case, the coefficients are numerically evaluated from samples. The Fourier expansion allows compact representation of the whole ARS in the form of a weighted sum of Fourier coefficients as the one in Equation (Equation 6). The advantages of this formulation emerge in the rotation estimation method presented in Section 3.2.

The main property of ARS lies in its invariance to translation t and angular shift. If translation t and rotation R with angle δ are applied to a point set represented by density function f(r), then the spectrum of the transformed point set satisfies the equation
(11)$$\begin{array}{c} \begin{array}{cc} & S\left[f(R(\delta )\;r+t)\right]\left(\theta \right)=S\left[f(r)\right]\left(\theta +\delta \right)\;. \end{array} \end{array}$$

Thus, the spectrum of a transformed point cloud is a shifted copy of the spectrum of the original point cloud, where the shift corresponds to the rotation angle δ. Rotation can be estimated using a proper metric and procedure for comparing spectra. In the next section, we show how this important property can be exploited to achieve this goal.

### 3.2. Registration Algorithm

ARS can be effectively applied to the estimation of the rigid transformation between two point clouds that represent the same scene from different viewpoints. Rotation is generally the more difficult part of the registration problem, and ARS translation-invariance allows decoupling of rotation estimation from translation estimation. As previously stated, when there is a rotation with angle δ between a source and a target point set represented by density functions, respectively, $f_{S}\left(r\right)$ and $f_{T}\left(r\right)$, spectrum $S\left[f_{S}\right]$ is the shifted copy of $S\left[f_{T}\right]$. The shift angle can be computed by searching the maximum of the following correlation function,
(12)$$C\left[f_{S},f_{T}\right]\left(\delta \right)=\frac{1}{\pi }\int_{0}^{\pi }S\left[f_{S}\right]\left(\theta +\delta \right)\;S\left[f_{T}\right]\left(\theta \right)\;d\theta ,$$
between the source and destination spectra. Since each ARS can be represented as a Fourier series, the correlation function is elegantly expressed in the form of convolution. The global maximum $\delta^{*}$ of $C[f_{S},f_{T}]$ can be efficiently found through a branch-and-bound procedure on the angular domain. More details can be found in [2,3]. It can be observed that, since ARS is π-periodic, the real rotation angle is either δ or δ+π. The assessment of translation enables disambiguation between the two candidate values of rotation.

To complete global registration, a branch-and-bound procedure inspired by [42] has been chosen. The objective function to be maximized is the number of overlapping point pairs between destination and the translated source point clouds. A point pair is overlapping if the distance in it is less than tolerance ε. Given a closed box $B\subset R^{2}$, the lower and upper bounds of the number of matching pairs are estimated. The lower bound is computed by counting the number of source points with a corresponding destination point belonging to the box B centered on the source point. The upper bound excludes from this counting the points clearly without matching. The translation is estimated as the center of the optimal box $B_{opt}$ which has the largest number of inliers.

## 4. Cud-ARS

### 4.1. Parallelization Setup and Enhancement

The ARS computational complexity is dominated by the evaluation of a spectrum kernel for each possible pair of points, as it is clear from Equation (Equation 6). Since $\psi_{ij}\left(\theta \right)=\psi_{ji}\left(\theta \right)$, the final spectrum does not depend on the processing order of the points with indices *i* and *j*. Conventionally, the point with an index (say, *i*) in the external loop is called *source point*, and the one with an index (say, *j*) in the nested loop is the *destination point*. The simple idea behind the GPU enhancement of isotropic ARS is to execute the largest number of computations in parallel.

In order to maximize computational throughput across the GPU kernels and to distribute computation in an efficient manner by doing so, a virtual grid-like structure to compute ARS spectra has been introduced. The goal has been to keep the number of threads a power of 2, starting from a minimum of 32. To preserve the square shape of the grid, its last cells still perform computation on padding data as part of ARS spectra computation pipeline, even though the padding data are not related to real pairs of points.

Parameter max_chunk_size corresponds to the maximum number of points taken as input into one Cud-ARS processing CUDA grid. If the size of the input source or destination data is greater than max_chunk_size, the Cud-ARS coefficient update is iterated until all the source-to-destination point comparisons are processed. This step is necessary as the internal memory of modern GPUs is rather large, but still finite. An additional check is performed to avoid Cud-ARS coefficient computation steps with small data chunks. When chunks of data slightly surpass the maximum size allowed, but they still fit the GPU memory, they are processed in one step to avoid the additional iteration that would slow down the whole pipeline (especially as less CPU-GPU transfers with high throughput are more efficient than multiple transfers with less data).

Cud-ARS is implemented as a three-step procedure. First, an indexed table of ARS coefficients is computed. The tidth element of this table corresponds to the evaluation of ARS on points with indices *i*, *j*, with i,j computed as explained in the getIJfromTID() method presented in Algorithm 1.   
**Algorithm** **1** Obtain I and J from TIDnId←n−1; //indices vary between 0 and n−1tid_tmp←tid**while** 
tid_tmp>=0 
**do**  tid_tmp−=(nId−i);  i←i+1;**end while****return** 
i←i−1;nId←n−1; //indices vary between 0 and n−1tid_tmp←tid**while** 
i>0 
**do**  tid_tmp−=(nId−i);  i←i−1;**end while****return** 
j←tid+1;

The tid indexing has been introduced in order to avoid excessive memory usage for storing useless computation outputs. As a matter of fact, the ARS coefficient matrix stores only the evaluation of ARS between points corresponding to non-null elements of the *strictly triangular* cost/matching matrix of the two datasets. Outputs *i* and *j* of Algorithm 1 correspond to the couple of point indices from source and, respectively, destination sets to be processed.

The most significant and computationally expensive step of this first part of the algorithm is the ARS kernel computation, which is on its own composed of two steps: an evaluation of the *PNEBI* (Product of Negative Exponential and Bessel functions on the first kind) which is defined as
(13)$$\begin{array}{cc} & PNEBI(k,x)=2.0*\exp(-x)*besseli(k,x), \end{array}$$
where besseli(k,x) is the modified Bessel function of the first kind of order *k*. The evaluation of besseli(k,x) is based on recurrence. Hence, it is convenient to evaluate all coefficients for k=0,…,arsOrder and to store them into vector pnebis. Said vector is used to update the coefficient matrix, as illustrated in the discussion of Algorithm 2.   
**Algorithm** **2** ARS Coefficient Downward Updatecoeffs=coeffsMat=matrix.empty()$factor\leftarrow weight=\frac{1.0}{\left(numPts^{2}\right)*\sqrt{\left(4*\pi *sigma^{2}\right)}}$rowIdx←tid=TODOfirstIdx=rowIdx∗ncols+0$\mathrm{coeffs}\left[firstIdx\right]+=0.5*factor*pnebis_{0}$$\mathrm{delta}\leftarrow means_{j}-means_{i}$phi←atan2(delta.y,delta.x)cth2←cos(2.0∗phi)sth2←sin(2.0∗phi)sgn←−1.0cth←cth2sth←sth2**for** 
k=1:fourierOrder 
**do**  evenIdx←rowIdx∗ncols+2k  oddIdx←rowIdx∗ncols+(2k+1)  coeffs[evenIdx]+=factor∗pnebis[k]∗sgn∗cth  coeffs[oddIdx]+=factor∗pnebis[k]∗sgn∗sth  sgn←−sgn  ctmp←cth2∗cth−sth2∗sth  stmp←sth2∗cth+cth2∗sth  cth←ctmp  sth←stmp**end for**

It can be noted that a large part of the computational load is due to the estimation of such 20÷30-sized vector for each pair of points to be evaluated with ARS. The aforementioned Algorithm 2 runs on the GPU in a for-stride loop guarded by the following instructions:(14)$$\begin{array}{c} \begin{array}{cc} & index=blockIdx.x*blockDim.x+threadIdx.x\\ & stride=blockDim.x*gridDim.x\\ & tot=gridDim.x*blockDim.x\\ & for(tid=index;\;tid<tot;\;tid+=stride) \end{array}, \end{array}$$
where index runs through a single block, while stride is supposedly the total number of threads in the grid. The goal is to fit as many coefficient computations as possible into one single grid.

Then, ARS computation for rotation estimation proceeds by summing the coefficients across each Fourier order, i.e., summing them along each (virtual) column of the coefficient matrix. However, due to the large number of rows to be summed for each column, it has appeared more profitable to subdivide this summing procedure into two steps: first, a partial column-wise sum of the coefficient matrix entries in chunks of consecutive rows (each having a fixed number of rows part_sum_numrows) is been computed; then, the sum of these partial sums is computed (still column-wise).

One last important consideration to be made is on how to approach large input datasets. The considerable amount of data needed for each thread of the kernels’ computation quickly fills the central memory of the GPUs, which for *general purpose* personal PCs rarely goes over 25 GB. Going even further into the exploitation of the separability of ARS coefficient parallel computation, the natural way for presented Cud-ARS to solve problems with input point sets with a size over ∼5000 points is subdivision of the datasets in chunks, and then the processing of each of the chunks separately. The 4096 value reported in Table 1 has been chosen as the appropriate maximum chunk size by empirical testing. The value of 256 representing the number of threads in each block has been selected in a similar fashion. Since kernel parallelization parameters vary depending on the input dataset chunk size, when the number of input points is greater than chunk_max_size, an iterative procedure resembling the subdivision into partial sums and total sums explained for computing ARS coefficients of each Fourier order is deployed. First, the parallelization parameters are updated according to input data chunk size. Then, the computation of ARS coefficients is summed for each combination of data chunks among source and destination point sets. Finally, it has to be noted that even during this process, the strictly triangular indexing for the ARS coefficient matrices is kept even when the need arises to evaluate ARS across multiple different chunk combinations, coming from source and destination point sets.

### 4.2. Full Registration and Mapping

While the precise estimation of rotation between two point clouds is an important task/primitive for plenty of robotics applications, it is even more important to integrate it in a more complete pipeline.

The generally accurate rotation estimation of ARS is used as the initial guess for the full pose estimation, paired with the translation estimation procedure presented in Section 3.2. The registration algorithm has been used to build maps by accumulating point clouds aligned with scan-to-scan matching. This solution, although simple, allows the appreciation of the effectiveness of the proposed pose estimation algorithm. Results are discussed in the next section.

## 5. Experiments

In order to evaluate Cud-ARS, multiple and diverse sets of experiments have been conducted. The first goal has been to assess whether Cud-ARS is able to obtain the same accuracy performances as ARS and other state-of-the-art methods on commonly used datasets (see Section 5.1). Second, the proposed methods for scan-matching based registration have been tested on real-world robotic simulations, assessing their ability in reconstructing the trajectory and their usability in building an occupancy grid map of the robot’s movements (see Section 5.2). An implementation of Cud-ARS is available in a public repository (available online at: https://github.com/ErnestF22/cudars/, accessed on 16 October 2023).

### 5.1. Cud-ARS Rotation Estimation

Cud-ARS evaluation has been performed on three datasets (namely *mpeg7*, *map* and *scan*) used also in previous works. These datasets contain a good range of the characteristics that can be found in robotic tasks with heterogeneous sensors and setup.

The *Mpeg7* datasets [43] are composed of images of more than 1000 different shapes that are sampled as point clouds. Three major transformations have been applied for testing the robustness of Cud-ARS:*Noise tests*. This transformation adds Gaussian noise with a given standard deviation σ to the points coordinates. The value of σ is varied in the interval of 0÷50, with the maximum dimension of a point set varying from 300 to 900.*Occlusion tests*. An occluded version of a point set is constructed by removing all points lying inside a randomly generated circle. The center of the circle is a randomly chosen point of the dataset, while the radius is proportional to the size of the point set. In particular, if the points are contained in a bounding box of size bx×by, the radius is equal to $\beta \sqrt{b_{x}b_{y}}$, where 0≤β≤1 is the occlusion rate. Occlusion rate β is varied up to 50%.*Rand tests*. This transformation adds $\gamma \;n_{in}$ random points to an input point set of $n_{in}$ points, where γ is the random point rate. The random points are uniformly distributed on a circle centered on the point set’s mean point, and with a radius double the size of bounding box. The maximum value of the random point rate γ used in the tests is 300, i.e., random points are at most three times the number of shape points.

The different transformations applied to assess the robustness of the rotation estimation are further explained in [2]. The *map* dataset is composed of occupancy grid maps obtained using the Cartographer [44] ROS tool on laser scans acquired at the University of Parma (available online: https://www.ce.unipr.it/~rizzini/papers/datasets/VLP-16_Unipr_DepartmentHall_dataset_20171102/, accessed on 16 October 2023), and on a public Deutsche Museum dataset (available online: https://google-cartographer-ros.readthedocs.io/en/latest/data.html, accessed on 16 October 2023). The *scan* dataset is made of laser scans traditionally used within the SLAM community, named after the place of acquisition: *fr-log*, *fr079*, *intel-lab* and *mit-csail* (URL to the *scan* dataset can be found in the Data Availability section). Each of these datasets contains about 5000 scans.

As expected, experiments on all these common robotic datasets show the same results for pose estimation as isotropic ARS, but with a substantial improvement in terms of speed. In these tests, Cud-ARS is compared against two previous versions of ARS (CPU Isotropic and Anisoptropic), and the Hough Spectrum [23] from Censi et al.

Results are shown in Figure 1, Figure 2 and Figure 3. The speed-up of the newly introduced Cud-ARS is easily noticeable across all experiments. The same can be said for the limited growth in execution time when other algorithms heavily slow down instead. It can be observed that the one algorithm performing similarly to Cud-ARS is Hough Spectrum (HS). While HS slightly outperforms Cud-ARS on some tests, we can still see that their speed is always very similar, and that they are constantly much faster than their counterparts. Cud-ARS achieves the same accuracy and precision as the original Isotropic ARS. This happens because, despite performing most of the computation in parallel on the GPU, Cud-ARS computes the same exact spectrum values in a more efficient way. Furthermore, the variance in terms of mean execution time is much higher for previous versions of ARS, which means that the newly introduced Cud-ARS can see a great increase in the number of potential applications. As a matter of fact, its performance is reasonably constant when dealing with diverse types of datasets (for example, in terms of the number of input points) whose elaboration may have previously required an excessive amount of time for online processing.

### 5.2. Registration and Mapping

In general, ARS registration has been able to keep track of the trajectories, even if working more similarly to a scan matcher rather than a classic registration method. To attest this, the proposed registration pipeline has been tested using a dataset acquired by a Pioneer 3DX robot in the main hallway of the Department of Engineering and Architecture at the University of Parma, which consists of a more than 100 m long corridor with branches and tables. The measurements included in the dataset are the robot odometry, the laser scans collected by Sick LMS100, and the point clouds collected by multilayer LIDAR Velodyne VLP16. In particular, we use two sequences called *uniprdia_0* and *uniprdia_1*.

The proposed ARS registration algorithm has been compared with Cartographer [44] and Hector SLAM (briefly Hector) [45]. At each iteration, the estimation given by ARS is based only on the alignment of the current laser scan with the previous one, without any initial guess. Conversely, Hector and Cartographer are full mapping systems that align and merge each new scan with the current map. Moreover, Hector uses the initial guess provided by the robot odometry and Cartographer also integrates the 3D measurements from the LIDAR. The goal of this tests is to display the robustness of ARS estimation, albeit based on scan-to-scan comparison.

Figure 4 and Figure 5 display the occupancy grid maps, as well as the estimated trajectories obtained with the three methods. The occupancy grid maps are obtained through online collection and merging of the aligned laser scans using Octomap [46], without removing inconsistencies, for a fair comparison of the three approaches. Even though it uses less data, ARS is able to estimate locally accurate trajectories. As it can be seen, though, in dataset *uniprdia_0*, after the robot performs a U-turn, the algorithm loses track of the real orientation. The loss in orientation might be due to a non sufficient rate of consecutive scan matching during the turn, or, even more trivially, to a simple badly acquired scan. More accurate results have been obtained on dataset *uniprdia_1*, where our method has been able to keep track of a more complex trajectory. It is well known that the absence of any kind of memory of previous states and maps while performing registration can lead to effects like this. However, these tests still show the stability of ARS’ scan-to-scan rotation and translation estimation, even if the need for adding a more complex mapping pipeline to the ARS project still appears necessary.

Table 2 reports the Average Translational Error (ATE) and Average Rotational Error (ARE) for Hector and ARS with respect to the trajectory of Cartographer (used as groundtruth) in the two sequences. As expected, ARS errors are larger, but significantly limited for a scan matching algorithm.

Another set of experiments has been conducted on the *scan* dataset. The goal of these experiments has been the one of assessing the ability to correctly estimate rigid transformations between subsequent scans in each dataset. Pose estimation tests have been separated into rotation and translation estimation, respectively. For rotation, the results obtained with the six methods are compared against the ground truth information contained in the datasets, as explained in [3]. Translation has been estimated after rotating the point clouds by an angle estimated through Isotropic ARS. The results are reported in Figure 6. ARS methods achieve an error on par with or lower than the other rotation estimation methods, while just a bit over the 1 cm scan resolution parameter when estimating translations. It has to be noted that several failed estimations are due to non-overlapping scans.

### 5.3. Discussion

The rotation estimation results presented in Section 5.1 and Section 5.2 show that the proposed method is constantly able to estimate rotation between pairs of point clouds with a <1[deg] accuracy. These results have been verified on multiple standard datasets, comprising images, occupancy grid maps, and planar laser scans. Cud-ARS achieves a speed up in the execution time of up to two orders of magnitude (i.e., up to 100×), rendering it at least on par with the grid-based HS in most tests. The scan-based mapping tests also assess the potential application of the proposed method to mapping tasks, even if one case shows that the incomplete representation provided by a pair of scans may lead the algorithm to losing track of the real robot orientation. However, this set of experiments has been designed to show the adequacy of the parallel rotation estimation and of the B&B-based translation algorithm as building blocks of a real-world mapping system.

## 6. Conclusions

This work has presented Cud-ARS, an algorithm able to perform fast and globally optimal registration on 2D point clouds. Cud-ARS is designed as a parallel algorithm for efficient computation of the Radon Spectrum. The original ARS has been reformulated in order to efficiently run on Nvidia GPUs. The ARS-based pipeline has been improved in its capability to perform full registration due to a B&B-based translation estimation method. Experiments conducted to compare and evaluate the presented method against state-of-the-art algorithms show the large improvements in computational speed of Cud-ARS, as well as the high accuracy of the method. The code is available on a public repository. The registration accuracy has been tested against more complex state-of-the-art mapping frameworks, and despite its current lack in place recognition capabilities, shows good potential even in real-world applications. Future work will include a more stable mapping pipeline, with online updates that will be performed on the GPU, in order to exploit the computational advantages even further.

## Figures and Tables

**Figure 1 sensors-23-08628-f001:**
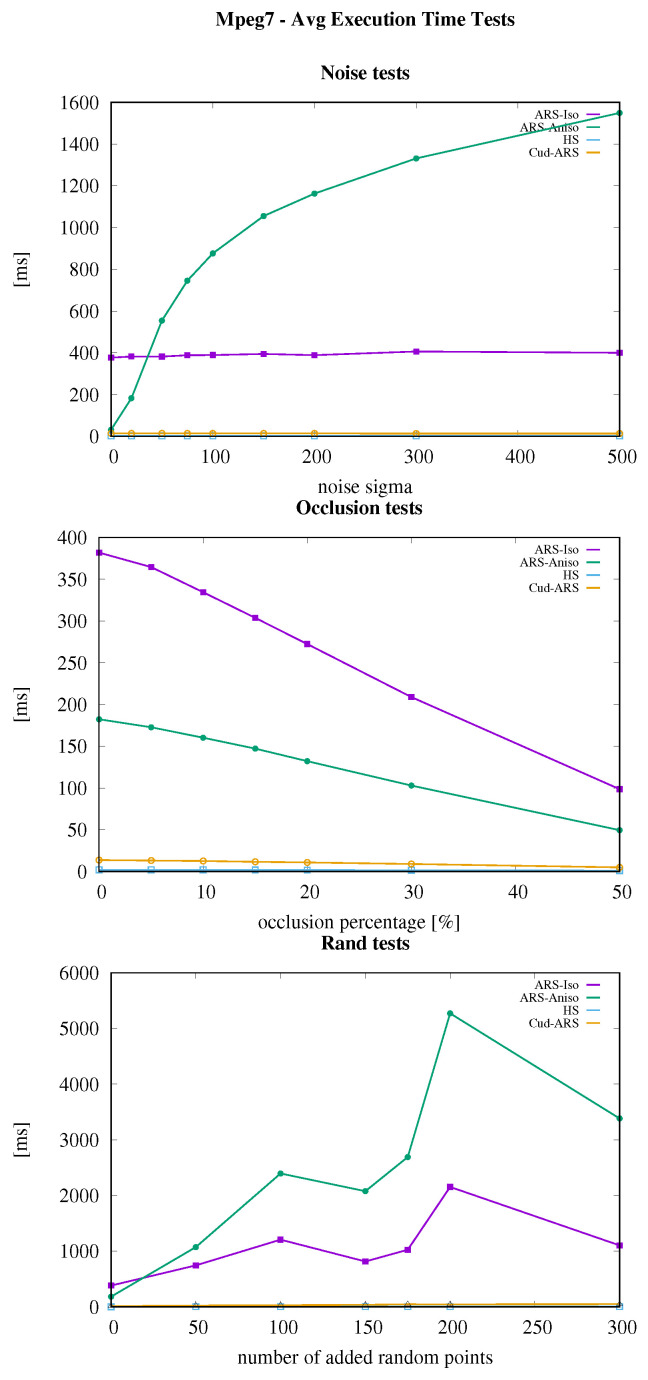
MPEG7 dataset execution time results.

**Figure 2 sensors-23-08628-f002:**
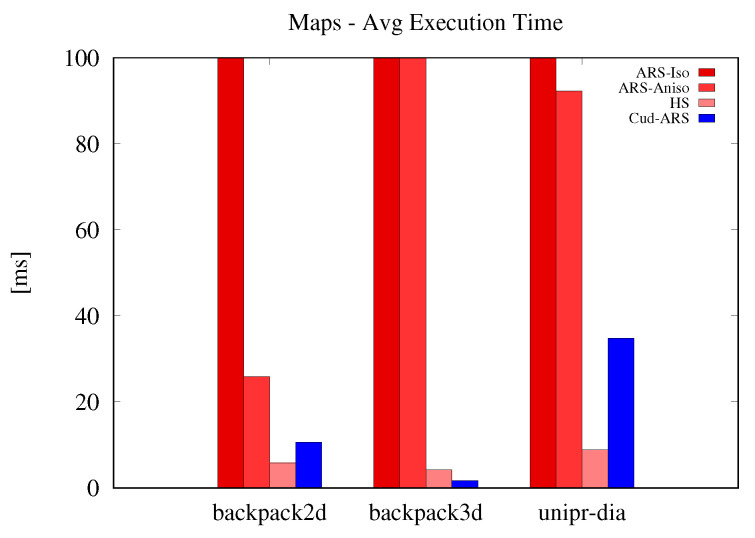
Maps dataset execution time results.

**Figure 3 sensors-23-08628-f003:**
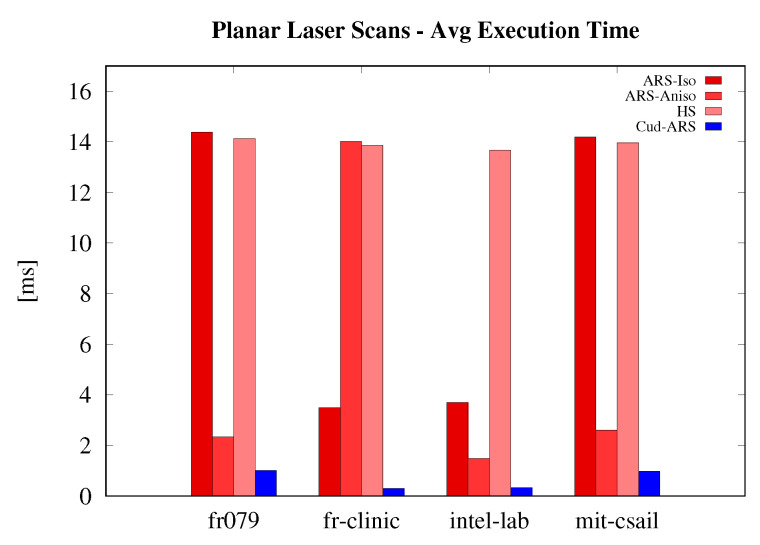
Scan dataset execution time results.

**Figure 4 sensors-23-08628-f004:**
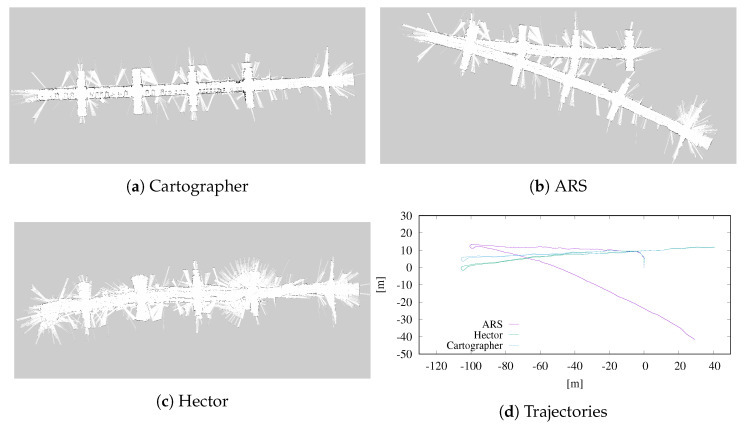
Occupancy grid maps and estimated trajectories of dataset *uniprdia_0* obtained using Cartographer, Hector SLAM and ARS-based registration. The occupancy grid maps are computed using Octomap that overlaps online raw laser scan data. Error propagation after the U-turn is visible.

**Figure 5 sensors-23-08628-f005:**
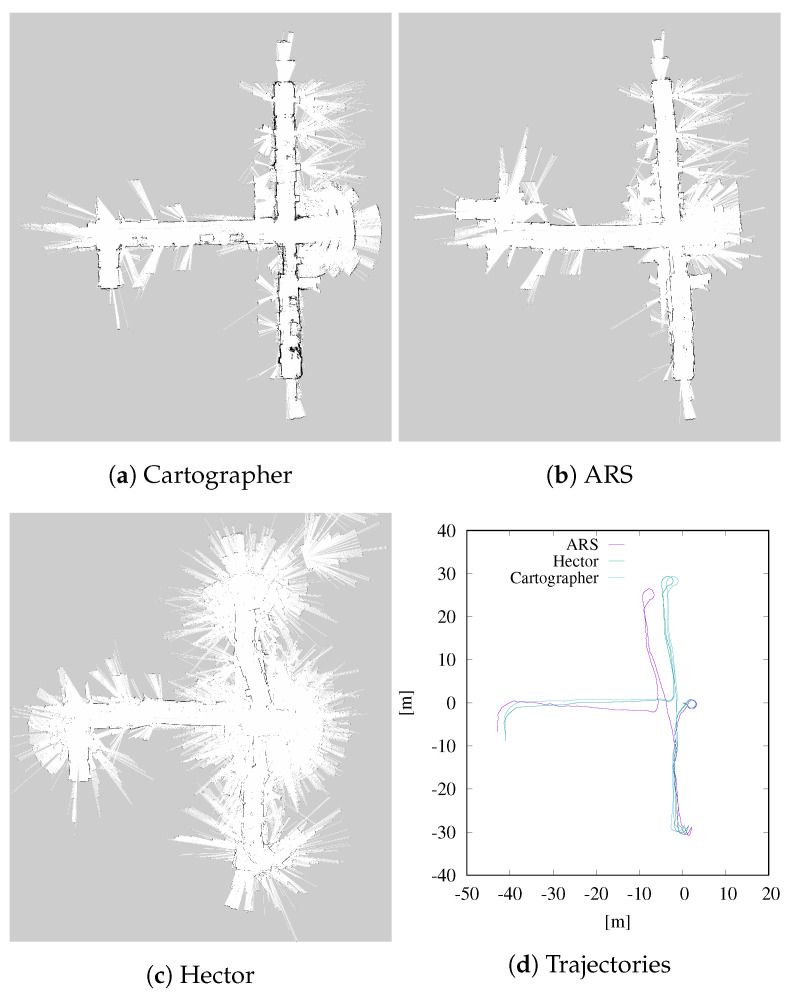
Occupancy grid maps and estimated trajectories of dataset *uniprdia_1* obtained using Cartographer, Hector SLAM and ARS-based registration. The occupancy grid maps are computed using Octomap that overlaps online raw laser scan data. Here, a more complex trajectory is kept track of with limited error.

**Figure 6 sensors-23-08628-f006:**
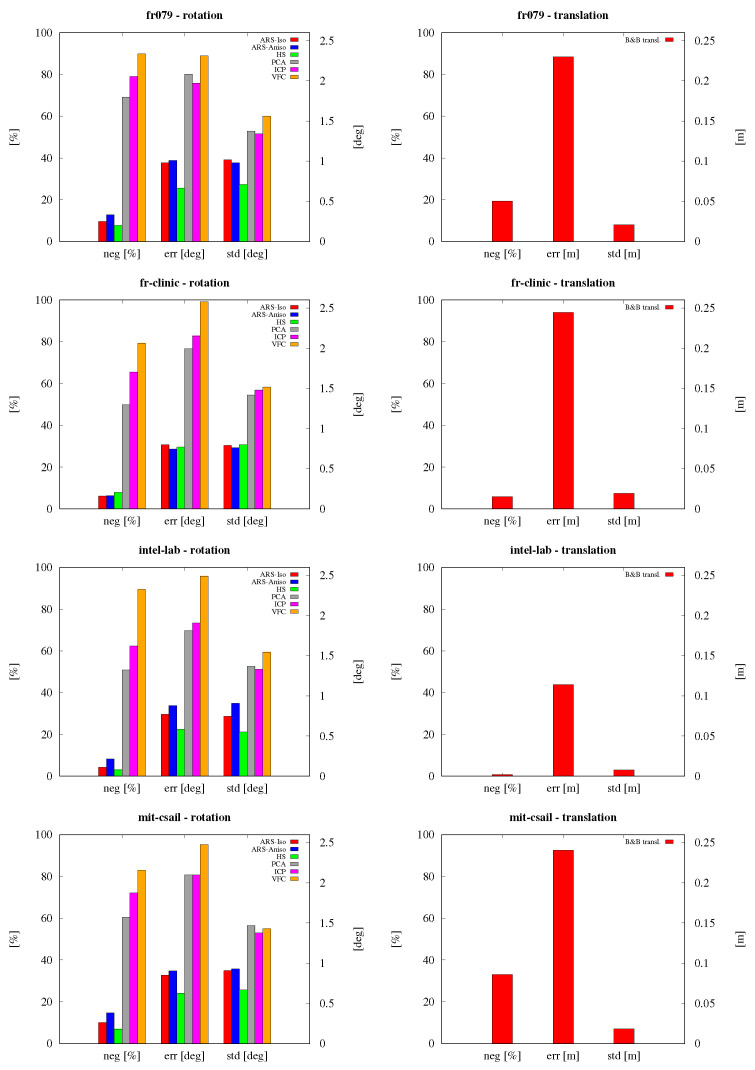
Registration accuracy in estimation of rotation (top) and of translation (bottom) on scan datasets *fr079*, *fr-clinic*, *intel-lab* and *mit-csail* (from left to right). For each set of tests, negative (failed) estimation percentage, average rotation error, standard deviation and translation mean error ($\left({}^{\circ }\right)$ and (m), respectively) are reported.

**Table 1 sensors-23-08628-t001:** Parameter configuration.

Description	Symbol	Value
ARS Fourier order	$n_{f}$	32
ARS stdev	$\sigma_{min}$	1.0 (mpeg7), 0.05 (maps)
ARS tolerance on B&B	Δθ	$0.5^{\circ }$
Coeff Matrix Rows	nrows	$\frac{num\_pts*(num\_pts-1)}{2}$
Coeff Matrix Cols	ncols	$2n_{f}+2$
Prlz Grid Size	grid_sz	nrows
Number of Blocks	blks	$\lfloor \frac{gridTotalSize}{pp.blockSz}\rfloor +1$
Number of Threads	threads	256
Coeff Matrix Tot Size	cffs_mat_tot_sz	grid_sz∗ncols
Max Chunk Size	max_chunk_size	4096

**Table 2 sensors-23-08628-t002:** Average Translational Error (ATE) and Average Rotational Error (ARE) obtained by Hector and ARS on the given sequences of datasets *uniprdia_0* and *uniprdia_1*.

Dataset	Length	Hector		ARS	
	(m)	ATE (%)	ARE ($10^{-2}{}^{\circ }/m$)	ATE (%)	ARE ($10^{-2}{}^{\circ }/m$)
*uniprdia_0*	262.28	3.87	7.18	19.78	31.66
*uniprdia_1*	180.34	3.14	6.66	11.06	32.58

## Data Availability

Code is available in public repository: https://github.com/ErnestF22/cudars/, accessed on 16 October 2023; *MPEG7* dataset is available at: https://www.ce.unipr.it/~rizzini/papers/datasets/mpeg7_dataset/, accessed on 16 October 2023; *scan* dataset is available at: https://www.ce.unipr.it/~rizzini/papers/datasets/laser2d_dataset/, accessed on 16 October 2023.

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
