# Peer review of "Accurate Global Point Cloud Registration Using GPU-Based Parallel Angular Radon Spectrum"

_sensors, 2023, doi:10.3390/s23208628_

Round 1

Reviewer 1 Report

The algorithms presented in the paper have been proved and published by the authors. So, the new application of the algorithms doesn't have any concerns. Therefore, I think that the paper is good for publication.

Author Response

Dear Reviewer,
Thank you for the appreciation showed for our manuscript.

Reviewer 2 Report

Dear colleagues, 

thank you for your contribution and development for a GPU-based Parallel Angular Radon Spectrum, the Cud-ARS, method, which is used for real-time environment mapping, point-cloud registration and robot localization. 

The paper is clear to read, the arguments can be followed, the literature review forms a fundament for the research that has been done and the results, which have been proven in a benchmark with Cartographer and Hector, are plausible. 

I do not have any concerns and issues for modification. 

Congratulation for your paper, 

best regards

Author Response

Dear Reviewer,
Thank you for the appreciation showed for our manuscript.
Best regards

Reviewer 3 Report

The algorithm is implemented in the Nvidia CUDA platform. The authors present the theoretical background and test results using several databases.

The article is technically sound. However, there are some aspects that need to be attended to improve its readability:

1.      Provide the reference for mpeg map and scan databases.

2.      The proposed Cud-ARS algorithm is compared against ARS Isotropic, anisotropic, and the Hough Spectrum (HS). Authors present results in Figure 1 regarding noise, occlusions “rand tests”. However, it is unclear what type of noise occlusions or rand tests were involved, please elaborate more, and describe better the scenarios.

3.      In Figure 1, it is noticeable that HS has similar performance as Cud-ARS and authors claim that “at no cost in terms of accuracy and precision” (page 9, line 275-276). Give a reference and/or explain this claim.

4.      In page 13, line 297 authors claim “The goal of such comparison is to display the robustness of ARS estimation” however when looking at the rotational and translational errors with Hector Slam in Table 2, the ARS shows higher errors. Please clarify how this robustness is achieved.

Author Response

Dear Reviewer,
Thank you for your consideration of our manuscript and the valuable suggestions. We addressed them in the following way:
1. References for the mpeg7 dataset were already present the Data Availability section. We added a reference for the scan dataset in the Data Availability section. We added footnotes with the URLs in section Experiments for dataset maps.
2. We illustrated the three transformations applied to the mpeg7 dataset in a bullet list in section 5.1.
3. Cud-ARS achieves the same accuracy and precision as the original Isotropic ARS. Despite reorganizing the computation in order to perform most of it in parallel on the GPU, Cud-ARS computes the same spectrum values in a more efficient way. We reformulated the sentence you mentioned in order to explain our claim.
4. This observation refers to the estimation achieved by Cud-ARS on scan-to-scan matching without initial guess. Hector SLAM exploits odometry data for registration and, moreover, compares the current scan with a local map that accumulates multiple scans. This is why we can claim that Cud-ARS's performance is robust, although the results reported in table II are inferior to those of Hector SLAM.

Reviewer 4 Report

In this paper, the authors present an efficient variant of ARS algorithm for 2D registration designed for parallel execution of the most computationally expensive steps on GPUs. The research has certain innovation, but there are also some details in the manuscript. I give suggestions for major repairs.

1. The abstract part should be further enriched, and specific digital results and analysis are needed.

2. In Page8,the manuscript format needs to be optimized.

3.In Page14,the font of  picture 6 needs to be larger, and two pictures can be considered to put a line.

4. The conclusion part may not be summarized in place, too little expression, please expand appropriately.

5. Please further standardize the format of references, especially some abbreviations.

Author Response

Dear Reviewer,
Thank you for your consideration of our manuscript and the valuable suggestions. We addressed them in the following way:
1. We extended the abstract by enriching the description of the main contributions and results of our work. We have improved the discussion of results in the Experiments section.
2. Unfortunately, the final size and format of the manuscript will be outcome of the editorial process. We will make sure to improve the formatting before publication.
3. We rearranged the graphs in figure 6 in order to make them larger and more readable.
4. We expanded the conclusion section with additional comments about the contributions and results.
5. We removed some "In proc. of" repetitions and we have made the abbreviations more uniform.

Round 2

Reviewer 4 Report

Thanks to the author's careful revision, and modify the following details, I can give the conclusion of the accept after minor revision.

1.The summary does not see specific results data, need to be clear.

2.Please increase the readability of fig.3 and fig.4. Now the fig.(a-c) in the figure does not see the legend, and the details in the figure are not clear, so it is not easy for people to intuitively understand.

3.The result analysis in the paper needs to be strengthened. Usually, the result and discussion part are written separately in the paper, and the author is asked to increase the discussion and analysis of the result.

Author Response

1. Thanks for the suggestion. We extended the abstract, also adding more comments on the results in the process.
2. Thanks for noting this detail. We believe that the reviewer was referring to figures 4 and 5. We increased the font size of the legends, as well as the size of the subfigures. We hope that this is sufficient to provide the additional clarity required by readers.
3. Thanks for the suggestion. We added subsection 5.3 where we recap and discuss the results more extensively.